A comprehensive analysis for associations between multiple microRNAs and prognosis of osteosarcoma patients

http://orcid.org/0000-0003-3281-7544 Yang Wen 1 2
http://orcid.org/0000-0003-3281-7544 Qi Yu-bin 3
Si Meng 1
Hou Yong 1
Nie Lin 1 nielinforest@163.com
1 Department of Orthopaedics, Qilu Hospital of Shandong University , Jinan, Shandong Province , China
2 Department of Spinal Surgery, Heze Municipal Hospital , Heze, Shandong Province , China
3 Department of Orthopaedics, Shandong Provincial Qianfoshan Hospital , Jinan, Shandong Province , China
Zhan Cheng
Electronic publication date: 2020 Jan 20
Publication date: 2020
Volume: 8
Electronic Location ID: e8389
Received 2019 Sep 4; Accepted 2019 Dec 13
Copyright: © 2020 Yang et al.
Copyright year: 2020
Copyright holder: Yang et al.
License: This is an open access article distributed under the terms of the Creative Commons Attribution License, which permits unrestricted use, distribution, reproduction and adaptation in any medium and for any purpose provided that it is properly attributed. For attribution, the original author(s), title, publication source (PeerJ) and either DOI or URL of the article must be cited.
License URL: https://creativecommons.org/licenses/by/4.0/

Keywords: MicroRNAs, Survival, Kaplan–Meier analysis, Multivariate Cox regression analysis

Funding: The authors received no funding for this work.

==============================
Background

Osteosarcoma (OS) is the most common malignant primary bone tumor occurring in children and young adults, which occupies the second important cause of tumor-associated deaths among children and young adults. Recent studies have demonstrated that many microRNAs (miRNAs) have abnormal expression in OS, and can function as prognostic factors of OS patients. However, no previous studies have comprehensively analyzed the relationship between multiple miRNAs and prognosis of OS patients.

Methods

A total of 63 OS patients were retrospectively enrolled. The clinical characteristics were collected, and the expression levels of miRNA-21, miRNA-30c, miRNA-34a, miRNA-101, miRNA-133a, miRNA-214, miRNA-218, miRNA-433 and miRNA-539 in tumor tissues were measured through quantitative real-time polymerasechain reaction. Kaplan–Meier analysis was used to perform univariate survival analysis, and Cox regression model was used to perform multivariate survival analysis which included the variables with P < 0.1 in univariate survival analysis.

Results

The cumulative survival for 1, 2 and 5 years was 90.48%, 68.25% and 38.10%, respectively, and mean survival time was (45.39 ± 3.60) months (95% CI [38.34–52.45]). Kaplan–Meier analysis demonstrated that TNM stage, metastasis or recurrence, miRNA-21, miRNA-214, miRNA-34a, miRNA-133a and miRNA-539 were correlated with cum survival, but gender, age, tumor diameter, differentiation, miRNA-30c, miRNA-433, miRNA-101 and miRNA-218 were not. Multivariate survival analysis demonstrated that miRNA-21 (hazard ratio (HR): 3.457, 95% CI [2.165–11.518]), miRNA (HR: 3.138, 95% CI [2.014–10.259]), miRNA-34a (HR: 0.452, 95% CI [0.202–0.915]), miRNA-133a (HR: 0.307, 95% CI [0.113–0.874]) and miRNA-539 (HR: 0.358, 95% CI [0.155–0.896]) were independent prognostic markers of OS patients after adjusting for TNM stage (HR: 2.893, 95% CI [1.496–8.125]), metastasis or recurrence (HR: 3.628, 95% CI [2.217–12.316]) and miRNA-30c (HR: 0.689, 95% CI [0.445–1.828]).

Conclusions

High expression of miRNA-21 and miRNA-214 and low expression of miRNA-34a, miRNA-133a and miRNA-539 were associated with poor prognosis of OS patients after adjusting for TNM stage, metastasis or recurrence and miRNA-30c.

Introduction

Osteosarcoma (OS) is the most common malignant primary bone tumor occurring in children and young adults, which occupies the second important cause of tumor-associated deaths among children and young adults (Mirabello, Troisi & Savage, 2009a, 2009b; Biermann et al., 2013; Yu et al., 2017). It is highly aggressive and occurs mainly in the proximal tibia, proximal humerus, and metaphyseal regions of the distal femur, with an incidence of 4.4 per million people around the world (Zhu et al., 2016). OS responds poorly to chemotherapy and the 5 year survival rate is still very low for OS patients with metastasis or recurrence (Hutanu et al., 2017; Zhou et al., 2016), although its prognosis has been improved gradually over the past 30 years (Rytting et al., 2000; Kunz et al., 2015). Therefore, it is crucial to identify new biomarkers that can exactly evaluate the prognosis of OS.

MicroRNAs (miRNAs) are a group of non-coding RNAs, which consist of 18–25 nucleotides (Ambros, 2004; Chang et al., 2016; Jamieson et al., 2012). They widely exist in animals, plants and even some viruses, and have an important role in post-transcriptional modulation of gene expression and gene silencing (Bartel, 2004; Hayes, Peruzzi & Lawler, 2014; Griffiths-Jones et al., 2008; Liu et al., 2017). Approximately 50% of miRNAs are confirmed to be associated with human tumorigenesis through directly targeting tumor suppressor genes or oncogenes (Li & Rana, 2014; Bracken, Scott & Goodall, 2016). MiRNAs are able to be circulated in body fluid, suggesting their potential as noninvasive markers (Bahrami et al., 2018). In OS, abnormal expression of miRNAs is involved in its occurrence and development. In addition, the expression of some miRNAs is associated with OS chemoresistance. Therfore, miRNAs have been widely applied in prediction of prognosis, detection of patients at early stages, and monitoring of the patients in response to chemotherapy. Studies have demonstrated that many miRNAs can function as prognostic factors of OS patients (Cheng et al., 2017; Zhang et al., 2015). Among them, miRNA-21, miRNA-30c, miRNA-34a, miRNA-101, miRNA-133a, miRNA-214, miRNA-218, miRNA-433 and miRNA-539 have been studied extensively and confirmed a potential association with the prognosis of OS patients. However, no previous studies have comprehensively analyzed the relationship between multiple miRNAs and prognosis of OS patients. There may be interactions among them. In this study, the expression levels of these nine miRNAs in tumor tissues of OS patients were measured through quantitative real-time PCR (qRT-PCR). Kaplan–Meier method was employed to determine the survival rate of OS patients, and long-rank test was employed to compare the survival rates between groups. Multivariate Cox regression analysis was finally performed to identify the independent prognostic factors with adjusting for confounders.

Materials and Methods

Patients

A total of 63 OS patients were retrospectively collected in Heze Municipal Hospital between January 2012 and January 2018. Surgery was performed in all of them, and tumor tissues and adjacent normal bone tissues were sampled. None of them received chemotherapy and radiotherapy before surgery. All tissue samples, obtained during surgery, were frozen immediately in liquid nitrogen and stored at −80 °C. The diagnosis and histological grading were determined with histopathological examination. This study received the approval of the ethic committee of Heze Municipal Hospital (20185261), and was performed according to the Declaration of Helsinki. All patients provided written informed consents.

Quantitative real-time PCR

Total RNA was extract from tumor tissues and adjacent normal bone tissues through miRNeasy kit (Qiagen, Hilden, Germany) in accordance with instructions of the manufacturer. The TaqMan miRNA assey kit (Applied Biosystems, Foster City, CA, USA) was used to quantitate the expression levels of miRNAs. Rotor Gene 6000 Real-Time PCR (Qiagen, Hilden, Germany) was used to perform Real-Time PCR with a TaqMan universal PCR master mix and an invitrogen kit. U6 was chosen as the reference gene, and the 2−ΔΔCt method was used to assess the relative expression levels of miRNAs. The primers of the included miRNAs and U6 were designed and chemosynthesized by Shanghai Jima Biotech Ltd. (Shanghai, China). The primers used were as follows: miRNA-21-3p: 5′-GCCACCACACCAGCTAATTT-3′ (forward) and 5′-CTGAAGTCGCCATGCAGATA-3′ (reverse); miRNA-30c-3p: 5′-GCCCAAGTGGTTCTGTGTTT-3′ (forward) and 5′-TCCATGGCAGAAGGAGTAAA-3′ (reverse); miRNA-34a-5p: 5′-TATGGCAGTGTCTTAGCTGGTTGT-3′ (forward) and 5′-GGCCAACCGCGAGAAGATG-3′ (reverse); miRNA-101-3p: 5′-GCCGAGTACAGTACTGTGA-3′ (forward) and 5′-CTCAACTGGTGTCGTGGA-3′ (reverse); miRNA-133a-5p: 5′-TGCTTTGCTAGAGCTGGTAAAATG-3′ (forward) and 5′-AGCTACAGCTGGTTGAAGGG-3′ (reverse); miRNA-214-3p: 5′-TGCAGTAGTGTCTTAGCTGGAATG-3′ (forward) and 5′-GGCTAACCGCGAGAAGTTT-3′ (reverse); miRNA-218-5p: 5′-GCGCTTGTGCTTGATCTAA-3′ (forward) and 5′-GTGCAGGGTCCGAGGT-3′ (reverse); miRNA-433-3p: 5′-GCTTTAGTGGTTCTGTGTGA-3′ (forward) and 5′-TCCGCGACAGAAGGAGTTTA-3′ (reverse); miRNA-539-3p: 5′-GCTTGTACACCAGCTAGTGC-3′ (forward) and 5′-CTTAGCTCGCCATGCAGAAG-3′ (reverse); and U6: 5′-GATCAAGGATGACACGCAAATTCG-3′ (forward) and 5′-GGCCAACCGCGAGAAGATG-3′ (reverse).

Statistical analysis

Statistical Analysis was conducted using the SPSS version 20.0 for Windows (SPSS Inc., Chicago, IL, USA). Kolmogorov–Smirnov test was used to determine the normality of quantitative data. Normal data were expressed as mean ± standard deviation, and non-normal data were expressed as median (interquartile range). Qualitative data were expressed as percentages or ratios (%). Kaplan–Meier analysis was used to perform univariate survival analysis, and Cox regression model was used to perform multivariate survival analysis which included the variables with P < 0.1 in univariate survival analysis. Significance was set at P < 0.05.

Results

General data

These 63 OS patients included 36 males and 27 females, and the median age of onset for them was 17 years with an interquartile range of 10 years. The other detailed clinical characteristics were demonstrated in Table 1. The follow-up was up to January 2019. The cumulative survival for 1, 2 and 5 years was 90.48%, 68.25% and 38.10%, respectively, and mean survival time was (45.39 ± 3.60) months (95% CI [38.34–52.45]).

Table 1 Clinical characteristics of OS patients.

Clinical characteristics	No. of patients	Percentages (%)	
Gender			
Male	36	57.14	
Female	27	42.86	
Age (years)			
≤25	55	87.30	
>25	8	12.70	
Tumor diameter (cm)			
≤5	37	58.73	
>5	26	41.27	
TNM stage			
I + II	25	39.68	
III + IV	38	60.32	
Metastasis or recurrence			
Yes	37	58.73	
No	26	41.27	
Differentiation			
Well and moderate	31	49.21	
Poor	32	50.79	

Expression levels of miRNAs in tumor tissues and adjacent normal bone tissues

According to the results of quantitative real-time polymerase chain reaction (Table 2; Fig. 1), the expression levels of miRNA-21, miRNA-214 and miRNA-433 were higher in tumor tissues than in adjacent normal bone tissues, and the expression levels of miRNA-30c, miRNA-34a, miRNA-101, miRNA-133a and miRNA-539 was lower in tumor tissues than in adjacent normal bone tissues, and the expression level of miRNA-218 was not statistically different.

Table 2 Expression levels of microRNAs in tumor tissues and adjacent normal bone tissues.

	microRNA-21	microRNA-214	microRNA-433	microRNA-30c	microRNA-34a	microRNA-101	microRNA-133a	microRNA-539	microRNA-218	
Tumor tissues	7.35 ± 2.96	6.12 ± 2.25	2.26 ± 1.34	3.93 ± 1.77	3.09 ± 0.94	3.16 ± 1.72	3.78 ± 2.17	2.35 ± 1.08	2.16 ± 1.07	
Adjacent normal bone tissues	3.14 ± 1.58	3.37 ± 1.49	1.17 ± 0.91	5.34 ± 1.32	5.24 ± 1.35	5.19 ± 2.74	11.89 ± 4.16	5.23 ± 1.84	2.31 ± 1.18	
t	9.959	8.088	5.341	−5.069	−10.374	−4.981	−13.719	−10.714	−0.747	
P	<0.05	<0.05	<0.05	<0.05	<0.05	<0.05	<0.05	<0.05	>0.05	

Figure 1 Expression levels of microRNAs in tumor tissues and adjacent normal bone tissues.

*P < 0.05 vs. tumor tissues.

Univariate survival analysis

The OS patients were divided into high expression group and low expression group according to the median expression levels of miRNAs. Kaplan–Meier analysis demonstrated that TNM stage (Fig. 2), metastasis or recurrence (Fig. 3), miRNA-21 (Fig. 4A), miRNA-214 (Fig. 4B), miRNA-34a (Fig. 4C), miRNA-133a (Fig. 4D) and miRNA-539 (Fig. 4E) were correlated with cum survival, but gender, age, tumor diameter, differentiation, miRNA-30c (Fig. 4F), miRNA-433, miRNA-101 and miRNA-218 were not. Median time of survival and log rank χ2 were demonstrated in Table 3.

Figure 2 Kaplan–Meier analysis of cumulative survival for TNM stage using Log Rank test.

Figure 3 Kaplan–Meier analysis of cumulative survival for metastasis or recurrence using Log Rank test.

Figure 4 Kaplan–Meier analysis of cumulative survival for microRNAs using Log Rank test.

(A) miRNA-21, (B) miRNA-214, (C) miRNA-34a, (D) miRNA-133a, (E) miRNA-539 and (F) miRNA-30c.

Table 3 Median time of survival and log rank χ2 for the K–M survival plots.

		No. of patients	Median time of survival (months)	Log rank χ2	P	
TNM stage	I + II	25	39.67 ± 4.43	4.199	0.040	
III + IV	38	49.15 ± 5.14	
Metastasis or recurrence	Yes	37	32.72 ± 3.85	28.970	<0.001	
No	26	63.42 ± 6.29	
microRNA-21	Low expression	24	61.75 ± 5.60	11.847	0.001	
High expression	39	35.32 ± 4.25	
microRNA-214	Low expression	26	58.24 ± 6.17	7.338	0.007	
High expression	37	36.36 ± 4.28	
microRNA-34a	Low expression	33	35.58 ± 4.22	5.372	0.020	
High expression	30	56.18 ± 5.87	
microRNA-133a	Low expression	42	36.35 ± 4.38	16.258	<0.001	
High expression	21	63.47 ± 5.89	
microRNA-539	Low expression	34	35.27 ± 4.13	7.390	0.007	
High expression	29	57.26 ± 6.07	
microRNA-30c	Low expression	32	42.41 ± 4.72	3.378	0.066	
High expression	31	48.47 ± 5.06	

Multivariate survival analysis

TNM stage, metastasis or recurrence, miRNA-21, miRNA-214, miRNA-30c, miRNA-34a, miRNA-133a and miRNA-539 were included in Cox proportional hazards model. According to the results of multivariate survival analysis (Table 4), miRNA-21 (hazard ratio (HR): 3.457, 95% CI [2.165–11.518]), miRNA-214 (HR: 3.138, 95% CI [2.014–10.259]), miRNA-34a (HR: 0.452, 95% CI [0.202–0.915]), miRNA-133a (HR: 0.307, 95% CI [0.113–0.874]) and miRNA-539 (HR: 0.358, 95% CI [0.155–0.896]) were independent prognostic markers of OS patients after adjusting for TNM stage (HR: 2.893, 95% CI [1.496–8.125]), metastasis or recurrence (HR: 3.628, 95% CI [2.217–12.316]) and miRNA-30c (HR: 0.689, 95% CI [0.445–1.828]). In other words, high expression of miRNA-21 and miRNA-214 and low expression of miRNA-34a, miRNA-133a and miRNA-539 were associated with poor prognosis of OS patients.

Table 4 Results of Cox proportional hazards model.

	Regression coefficient	Standard error	Wald χ2	Hazard ratio	95% Confidence interval	P	
microRNA-21	1.107	0.465	5.923	3.457	2.165–11.518	0.013	
microRNA-214	1.058	0.446	5.642	3.138	2.014–10.259	0.017	
microRNA-34a	−0.835	0.371	5.148	0.452	0.202–0.915	0.021	
microRNA-133a	−0.946	0.382	6.137	0.307	0.113–0.874	0.011	
microRNA-539	−0.887	0.369	5.474	0.358	0.155–0.896	0.018	
TNM stage	0.953	0.392	5.016	2.893	1.496–8.125	0.024	
Metastasis or recurrence	1.154	0.458	6.529	3.628	2.217–12.316	0.007	
microRNA-30c	−0.738	0.426	3.045	0.689	0.445–1.828	0.074	

Discussion

The prognosis of OS patients has been significantly improved with the development of multiple chemotherapy regimens. However, OS patients receiving the same treatment often demonstrate different clinical outcomes, suggesting an urgent need for developing reliable prognostic biomarkers to improve the prognosis of OS patients. MiRNAs modulate protein expression through regulating the degradation and translation of mRNAs at post-transcriptional level (Chang et al., 2016; Jamieson et al., 2012). They play a critical role in various biological processes which are involved in the development and progression of tumors, including proliferation, apoptosis, differentiation and metastasis (Hayes, Peruzzi & Lawler, 2014; Ebert & Sharp, 2012; Rogers & Chen, 2013; Liu et al., 2012).

Additionally, they are very stable and easily detected in the blood and tissues (Gilad et al., 2008). Therefore, plenty of miRNAs are employed as new biomarkers for the diagnosis and prognosis of tumors. Regarding to OS, a variety of miRNAs has been reported to be associated with its prognosis. Kim et al. demonstrated that the pooled HR was 1.40 (95% CI [1.01–1.94]) for OS patients with lower expression miRNAs, and proposed that miRNAs with increased expression should also be investigated for their effects on the prognosis of OS patients. Additionally, the expression of some miRNAs is associated with OS chemoresistance (Xie et al., 2018). In our study, the nine miRNAs, having been studied widely, were chosen as research targets. Our results demonstrated that miRNA-21, miRNA-214, miRNA-34a, miRNA-133a and miRNA-539 were independently associated with the prognosis of OS patients.

MiRNA-21 has been confirmed to act as tumor oncogene in many types of tumors. For OS, it may regulate the proliferation, invasion and metastasis of OS cells through directly targeting PTEN and RECK (Ziyan et al., 2011; Lv, Hao & Tu, 2016). Li et al. (2018) demonstrated that the elevated expression of miRNA-21 might lead to elevated expression of the proteins in the PI3K/AKT signaling pathway and decreased expression of PTEN, which was associated with the increased invasiveness of OS cells. Hu et al. (2018) indicated that inhibition of miRNA-21 might reduce the proliferation of OS cells through modulating the TGF-β1 signaling pathway and targeting PTEN. Additionally, miRNA-21 might decrease the anti-tumor effect of cisplatin through modulating the expression of Bcl-2 (Ziyan & Yang, 2016). Our results demonstrated that high expression of miRNA-21 was independently associated with poor pognosis of OS patients with a HR of 3.457 (95% CI [2.165–11.518]). MiRNA-214 may act as either a tumor suppressor gene or an oncogene. For OS, the elevated expression of miRNA-214 is associated with enhanced invasion and proliferation of OS cells through modulating the expression of LZTS1 (Xu & Wang, 2014). However, Rehei et al. (2018) found that the expression of miRNA-214 was negatively associated with the expression of TRAF3 in OS tissues, and over-expression of miRNA-214 could inhibit the invasion and metastasis of OS cells through targeting TRAF3. Our results demonstrated that high expression of miRNA-214 was independently associated with poor prognosis of OS patients with a HR of 3.138 (95% CI [2.014–10.259]).

MiRNA-34a has various target genes which play important roles in biological function of OS cells, such as Fag1, Wnt, p53 and Notch (Wu et al., 2013; Yan et al., 2012). Gang et al. (2017) demonstrated that miRNA-34a was correlated with the apoptosis, proliferation and adhesion of OS cells, and could function as a new tumor suppressor gene by reducing the expression of DUSP1. Zhang et al. (2018) proved that miRNA-34a was a crucial regulator in the dedifferentiation of OS cells through modulating PAI-1-Sox2 axis. In addition, Wang et al. (2018) showed that down-modulated expression of miRNA-34a was a prognostic biomarker for poor prognosis of OS patients through a meta-analysis. Our results demonstrated that low expression of miRNA-34a was independently associated with poor prognosis of OS patients with a HR of 0.452 (95% CI [0.202–0.915]). MiRNA-133a has been proved to be a crucial modulator for osteogenesis, and have a key role in osteoblast differentiation (Bao et al., 2010). It can act as an antionco-miRNA or a tumor suppressor gene in the development and progression of tumors (Ji et al., 2013). It has been reported to be associated with many cancers, including esophagus cancer, bladder cancer and prostate cancer. The underlying mechanisms of pro-apoptotic function of miRNA-133a may be associated with the inhibition ofMcl-1 and Bcl-xL expression (Wang et al., 2010). Our results confirmed that low expression of miRNA-133a was independently associated with poor prognosis of OS patients with a HR of 0.307 (95% CI [0.113–0.874]). Few reports have investigated the biological functions of miRNA-539. Muthusamy et al. (2014) found that miRNA-539 could inhibit O-GlcNAcase expression. Wang et al. (2014) demonstrated that miRNA-539 was involved in the regulation of apoptosis and mitochondrial activity by means of targeting PHB2. The expression of miRNA-539 is down-regulated in thyroid cancer, and moreover, it has a suppressor role in the invasion and metastasis of thyroid cancer cells through targeting CARMA1 (Gu & Sun, 2015). Our results demonstrated that low expression of miRNA-539 was independently associated with poor prognosis of OS patients with a HR of 0.358 (95% CI [0.155–0.896]).

Conclusions

High expression of miRNA-21 and miRNA-214 and low expression of miRNA-34a, miRNA-133a and miRNA-539 were associated with poor prognosis of OS patients after adjusting for TNM stage, metastasis or recurrence and miRNA-30c.

Supplemental Information

Supplemental Information 1 Raw data for expression levels of MiRNAs in adjacent normal bone tissues.

Click here for additional data file.

Supplemental Information 2 Raw data for Kaplan-Meier analysis.

Click here for additional data file.

Additional Information and Declarations

Competing Interests

Author Contributions

Human Ethics

Data Availability

The authors declare that they have no competing interests.

Wen Yang performed the experiments, authored or reviewed drafts of the paper, and approved the final draft.

Yu-bin Qi performed the experiments, prepared figures and/or tables, data management, and approved the final draft.

Meng Si performed the experiments, prepared figures and/or tables, and approved the final draft.

Yong Hou performed the experiments, analyzed the data, prepared figures and/or tables, and approved the final draft.

Lin Nie conceived and designed the experiments, authored or reviewed drafts of the paper, and approved the final draft.

The following information was supplied relating to ethical approvals (i.e., approving body and any reference numbers):

This study received the approval of the ethic committee of Heze Municipal Hospital (20185261).

The following information was supplied regarding data availability:

All raw data are available in the Supplemental Files.

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
