# Peer review of "A comprehensive analysis for associations between multiple microRNAs and prognosis of osteosarcoma patients"

_PeerJ, doi:10.7717/peerj.8389_

## Round 0.1 · original submission · Major Revisions

Manuscript entitled "A comprehensive analysis for associations between multiple microRNAs and prognosis of osteosarcoma patients" which you submitted to PeerJ, has been reviewed. The reviewers have recommended publication pending major revisions. Therefore, I invite you to respond to the reviewers' comments at the bottom of this letter and revise your manuscript accordingly.

Reviewer 1 ·

Basic reporting

no comment

Experimental design

The authors should list the primer sequences of the included miRNAs and U6.
The authors should explain why they studied the included miRNAs and why they did not study other miRNAs. I think it is not possible to study miRNAs randomly and there are thousands of miRNAs in human. It is not enough that the authors just depended on the reports of previously published articles.
There are two types of mature miR-21 and please make it clear that miR-21-3p or miR-21-5p the authors used. Similarly, the authors should make it clear which type they studied about miR-214, miR-433, miR-30c, miR-34a, miR-101, miR-133a, miR-539, miR-218.

Validity of the findings

no comment

Additional comments

no comment

·

Basic reporting

Figures
- I recommend combining figures 3 through 8 into one figure.
- Please make table including the results of cox regression.

Literature references, sufficient filed background/context
- The authors cited Cheng D et al (MicroRNAs with prognostic significance in osteosarcoma~) and they focused on each microRNAs. Please cite Kim YH et al., 2017 and discuss the risk of pooled microRNAs in osteosarcoma patients. In addition, why don't you combine microRNAs to make stronger prognostic predictions or diagnostic markers?

- In the discussion, please discuss the results (microRNAs expression and prognostic significance) in detail. Now the authors focused on previous molecular studies.

Raw data shared
- If the authors share the qRT-PCR and paired clinical data, it will help to many researchers.

Experimental design

Methods
- Please describe in detail.
- In survival curve, they might use the qRT-PCR data from tumour tissue. How about to use differences or fold change between tumour and paired normal tissues.
- In cox regression, please describe how they use expression data as categorical or continuous value.

Validity of the findings

- There is one cohort containing microRNAs and clincial data (GSE30934). By using the cohort, please identify the association between chemotherapy response and microRNA expression.

- If possible, the authors check the microRNAs function through microRNA transfection.

Additional comments

Thank you for giving a chance for reviewing the manuscript. The manuscript is very interesting, however, there are some scientific and technical issues to address.

Reviewer 3 ·

Basic reporting

Osteosarcoma (OS) mainly arises from the metaphysis of the long bones and it is the most common primary bone malignancy in adolescents and young adults. Despite advances in disease management and treatment, the long-term prognosis of patients with OS is still poor.
However, the authors should clearly explain
1. What is the gap in the existing literature and why do they need to use miRNA as the prognostic markers?
2. What is the reason(s) for the authors only to look at the few miRNAs mentioned in this study? It is ideal and unbiased to have done transcriptome profiling of miRNA array for the patient samples they posses from which they could have bioinformatically narrowed-down to pin point which amongst the miRNAs are relevant to prognosis. If they have the tumor and control samples it would be a better study to do the RNA seq for miRNAs.

Few sentences were not clear:
1. In the abstract and result please use the full form for any abbreviation.
2. The meaning of the sentences in line 110 is not clear. Please elaborate.

Experimental design

1. It is unclear from this manuscript the gap in the existing literature and why do they need to use miRNA as the prognostic markers.
2. Please read Zhang et al 2015, Clinica Chimica ACTA where a list of miRNAs with their expression and relation OS are presented in table1. The premise of selectively choosing few miRNA in this study remains unclear.
3. The authors have performed qRT-PCR for OS and control samples to analyze the expression levels of the miRNAs. Could you please plot graphical plots to show the result along with the table you provided? Please plot ΔΔCt with standard deviation or any meaningful way.
4. If you have the details of these OS tumors as in table 1, could you also specify what stages of the OS tumors were used to perform qRT-PCR (was done by pooling all the stages/grades of OS together or there were separate samples per stage/grade of OS)? It is important to know the expression pattern over the progression of the disease. Provide elaborate explanation.
5. Also, could you provide evidence of the expression of miRNAs in the metastatic and recurrent tumor samples.
6. As a proof of principle, please use TCGA datasets OS patient samples to analyze and find whether the specific miRNAs are differentially regulated between OS and control samples in an independent dataset.
7. Method section is very poorly written. As I can understand that this is a bioinformatics-based manuscript, please provide detail description of all the computational methods used.
8. What is the purpose of selecting p-value <0.1 for multivariate analysis?
9. Each of the K-M survival plots should be indicative of no. of patients in each cohort and median time of survival. Very important please provide data as Tables for all the figures and keep the color code same all throughout (eg. Hig expression as green and low expression as blue maintain same color for all graphs).
10. What is the cut-off used to segregate low vs high miRNA expression and why?

Validity of the findings

Data analysis shows miRNA 21 and 214 (high expression) while 34a, 133a, 539 (low expression) are poor prognostic markers for OS.
Please refer to questions in the above panel.

Additional comments

The study is a short bioinformatics one. It could be potentially better with answers to the above mentioned questions. An independant study using the TCGA OS dataset will provide not only proof of principle to your study but also an in-depth overview of miRNAs in OS patient survival and prognosis.

---

## Round 0.2 · accepted · Accept

I am writing to inform you that your manuscript has been accepted for publication. Congratulations!

Reviewer 1 ·

Basic reporting

no comment

Experimental design

no comment

Validity of the findings

no comment

Additional comments

no comment

·

Basic reporting

The authors revised the manuscript very well.

Experimental design

The authors revised the manuscript very well in Experimental design.

Validity of the findings

The authors revised the manuscript very well.

Additional comments

The authors revised the manuscript very well.

Reviewer 3 ·

Basic reporting

Overall, the revised manuscript has certainly improved.
Authors successfully answered most of the questions, however they were unable to address few questions because of insufficient funds as the first author mentioned. Minor changes like spellings the authors can take care at the time of proof-read.
I do think the study is suitable for PeerJ journal submission.

Experimental design

Overall, the revised manuscript has certainly improved.
Authors successfully answered most of the questions, however they were unable to address few questions because of insufficient funds as the first author mentioned. Minor changes like spellings the authors can take care at the time of proof-read.
I do think the study is suitable for PeerJ journal submission.

Validity of the findings

Overall, the revised manuscript has certainly improved.
Authors successfully answered most of the questions, however they were unable to address few questions because of insufficient funds as the first author mentioned. Minor changes like spellings the authors can take care at the time of proof-read.
I do think the study is suitable for PeerJ journal submission.

Additional comments

Overall, the revised manuscript has certainly improved.
Authors successfully answered most of the questions, however they were unable to address few questions because of insufficient funds as the first author mentioned. Minor changes like spellings the authors can take care at the time of proof-read.
I do think the study is suitable for PeerJ journal submission.